# Suppression of Smoldering of Calcium Alginate Flame-Retardant Paper by Flame-Retardant Polyamide-66

**DOI:** 10.3390/polym13030430

**Published:** 2021-01-29

**Authors:** Kai Xu, Xing Tian, Ying Cao, Yaqi He, Yanzhi Xia, Fengyu Quan

**Affiliations:** 1State Key Laboratory of Bio-Fibers and Eco-Textiles, Collaborative Innovation Center of Marine Biobased Fiber and Ecological Textile Technology, Institute of Marine Biobased Materials, Qingdao University, Qingdao 266071, China; 2018020388@qdu.edu.cn (K.X.); xingtian1982@qdu.edu.cn (X.T.); xyz@qdu.edu.cn (Y.X.); 2College of Materials Science and Engineering, Qingdao University, Qingdao 266071, China; 2019020444@qdu.edu.cn (Y.C.); 2019204670@qdu.edu.cn (Y.H.)

**Keywords:** calcium alginate fiber, flame retardant polyamide-66 fiber, blending, flame-retardant paper, smoldering, flame retardancy

## Abstract

Calcium alginate (Ca-Alg) fibers are renewable fibers obtained from the ocean with essential flame retardancy, which have recently been applied as components of flame-retardant paper. However, the application of Ca-Alg fibers is limited because of their tendency to smolder. Therefore, composites papers were fabricated by blending using flame-retardant polyamide-66 (FR-PA), with a 5 wt% content of phosphorous flame retardant, which will form molten carbon during combustion. When the FR-PA content is 30% of the composite paper, FR-PA forms a compact carbon layer on the surface of the Ca-Alg fibers during combustion, which isolates the mass/heat transfer and effectively suppresses the smoldering of Ca-Alg. This consists of a condensed flame retardant mechanism. Furthermore, the combustion and thermal degradation behavior of paper were analyzed by cone calorimetry (CONE), TG and TG-IR. Ca-Alg in the composite paper decomposed and released CO_2_ before ignition, which delayed the ignition time. Simultaneously, the FR-PA contained in the composite paper effectively inhibited the combustion of volatile combustibles in the gas phase. Overall, FR-PA and Ca-Alg improve the thermal stability of the composite paper in different temperature regions under air atmosphere. Ca-Alg reduces the formation of aromatic products and NH_3_ in the composite paper under N_2_ atmosphere. Ca-Alg-based paper with excellent flame retardancy was successfully prepared.

## 1. Introduction

In modern society, wallpapers are becoming increasingly popular as decoration materials because of their variety of patterns that meet individual needs. However, most wallpaper consists of cellulose and lignin [1], which have great security risks, and their flame can easily spread along the paper, resulting in the loss of human life and property in the case of fire [2,3,4,5,6]. At present, flame-retardant paper is prepared by the impregnation, coating, or addition of flame retardants to the pulp. Halogen flame retardants have been gradually replaced by phosphorus and nitrogen-based flame retardants due to their environmental hazard. Phosphorus-based flame retardants can promote the formation of a carbon layer and the NH_3_ generated by nitrogen-based flame retardants can dilute the concentration of combustible gases. The influence of inorganic flame retardants (such as Al(OH)_3_ and Mg(OH)_2_) on the flame retardancy characteristics of products has been studied, and it was revealed that their incorporation affects the mechanical strength of the material [4,7,8,9,10,11,12]. In order to improve the flame retardancy of pulp fiber-based paper, Zhang et al. [4] synthesized cellulose nanofibril-based flame retardants by combining phosphorylated cellulose nanofibril and ionic bonding melamine. Mao et al. [13] prepared polyaniline-deposited functional cellulose paper by doping with organic sulfonic acid, which improved the oxygen index of cellulose paper. Pan et al. [14] prepared a zinc-coordinated multi-coated flame-retardant paper by layer-by-layer self-assembly, and the peak of heat release rate (pHRR) and total heat release (THR) values of treated paper were significantly reduced.

At present, products containing flame retardants are widely used in living and working places in items such as furniture, curtains, wallpaper, among others. Flame retardants are released from discarded products, causing severe environmental pollution [15]. Therefore, reducing the use of flame retardants is a direct way to reduce their harm to the environment. Calcium alginate (Ca-Alg) fiber is a renewable fiber derived from algae and prepared by the wet spinning of sodium alginate, which has the characteristics of inherent flame retardancy and potential bioactivity [16,17]. Zhang et al. [17] determined the flame-retardant mechanism of Ca-Alg fibers. In comparison with viscose fibers, which are also derived from biomass, the calcium ion can catalyze the degradation process of the alginate fiber and produce more residues containing carbonaceous char and carbonate. In addition, the molecular chain of the Ca-Alg fiber contains a large number of -COOH and -OH groups, which can be dehydrated to form cyclic lactide when heated. This dilutes the concentration of combustible gas and inhibits heat transfer [18,19,20]. Therefore, the application of Ca-Alg fibers to flame-retardant paper has a great application prospect. However, it was found that although Ca-Alg fibers cannot be ignited, smoldering limits their application [21]. Polyamide fibers have been widely used as a synthetic fiber in the military and civilian fields [22,23]. However, polyamides containing aliphatic components are often flammable and have a low carbon residue because they are easily transformed into volatile fuel [24]. At present, the main flame retardants used in polyamide fabrics are brominated ones, thiourea formaldehyde resin, and cyclic phosphoryl chloride derivatives [23,25,26,27]. Unfortunately, flame-retardant polyamide (FR-PA) fibers melt easily, which may cause secondary injuries such as scalding [28].

Blending is a low-cost and high-efficiency method of mixing two or more kinds of materials to maximize their respective advantages [29]. Studies have confirmed that the flame retardancy of materials can be improved by blending different components in a certain proportion, such as flame-retardant viscose/alginate fibers, flame-retardant cotton/alginate fibers, and flame-retardant vinylon/poly(m-phenylene isophthalamide) fibers, among others [21,29,30,31,32,33]. However, not all fibers can improve their flame retardancy by blending, but it depends on whether there is a condensed or gas phase interaction between the blended fiber components during combustion [32,34]. Zhang et al. [21] stated that Ca-Alg fibers are prone to smoldering due to their low initial degradation temperature, and proposed that the smoldering of Ca-Alg fibers can be inhibited by blending with flame-retardant viscose fibers. In the composite, the carbon in the flame-retardant viscose fibers can prevent heat transmission, further inhibiting the smoldering of the Ca-Alg fibers. In addition, the flame-retardant properties of polyamide/alginate (PA/Ca-Alg) non-woven fabrics have been studied, and it was revealed that the molten PA is restricted to the carbon residue area of Ca-Alg, which improves the flame-retardant properties of the non-woven fabrics [33]. Therefore, it can be hypothesized that molten PA forms a compact carbon layer that may inhibit the smoldering of the Ca-Alg fibers.

This article explores the interaction between Ca-Alg fibers and flame-retardant polyamide-66 (FR-PA) fibers during combustion, to explore their application in the field of flame-retardant paper. It is hypothesized that the smoldering of Ca-Alg fibers or the melting-drop of the FR-PA fibers can be inhibited by blending. It is also worth exploring whether the FR-PA/Ca-Alg composite paper can meet the flame-retardant requirements of flame-retardant paper, which, according to GB/T 14656-2009, is determined by an after-flame time lower than 5 s, an afterglow time lower than 60 s, and a damaged length lower than 115 mm. FR-PA/Ca-Alg composite papers were prepared by blending FR-PA pulp fibers with Ca-Alg pulp fibers. Furthermore, the flame retardancy and combustion behavior of the composite flame-retardant papers were studied, and the results were compared with commercial wood pulp paper

## 2. Materials and Methods

### 2.1. Materials

Ca-Alg fibers were obtained from Qingdao Yuanhai New Material Technology Co., Ltd. (Qingdao, China). FR-PA fibers were supplied by Huafeng Group Co., Ltd. (Wenzhou, China). The flame retardant contained in FR-PA is phosphorus-based, with a 5 wt% content, and the structural formula is shown in Appendix A. Bleaching Eucalyptus pulp was purchased from Shandong Daoxin New Material Co., Ltd. (Jinan, China). Deionized water was prepared by a reverse osmosis pure water equipment (Sihai Water Treatment Equipment Co., Ltd., Weifang, China).

### 2.2. Preparation of FR-PA/Ca-Alg Papers

The composite papers were prepared using the Rapid Kothen method [35,36], and the process is shown in Figure 1. First, the FR-PA or Ca-Alg fibers (500 g) and deionized water (30 L) were put into a Varley beater (AT-WD, Shandong Anmt Instrument Co., Ltd., Jinan, China) to make into pulp by beating the fibers for 2 h. Then, the obtained FR-PA and Ca-Alg pulps were weighed to prepare mixtures with a mass ratio of 0:100, 10:90, 20:80, 30:70, and 40:60 (*w*/*w*). The mixed pulp (10 g) and deionized water (2.5 L) was subsequently placed in a standard fiber dissociator (AT-XW, Shandong Anmt Instrument Co., Ltd.) at 1000 rpm for 20 min for dispersion. Then, the homogeneous slurry was quickly transferred to a sheet former (AT-CZ-3, Shandong Anmt Instrument Co., Ltd.) for papermaking. Finally, the obtained paper was dried in a hot-oven at 70 °C for 1 h. The thickness of the paper was 0.4 mm and the area was 314 cm^2^. The obtained papers were noted as Ca-Alg, FR-PA/Ca-Alg (10/90), FR-PA/Ca-Alg (20/80), FR-PA/Ca-Alg (30/70), and FR-PA/Ca-Alg (40/60). The preparation process of wood pulp paper is the same as that of composite paper. The composition of the prepared paper is shown in Appendix A, and the grammage of the prepared paper was approximately 305 ± 3 g/m^2^.

### 2.3. Characterization

#### 2.3.1. Physical Properties of the Prepared Paper

The Bendtsen roughness of the prepared paper was tested according to ISO 8791-2:1990, and its air permeability was analyzed according to GB/T 22819-2008. The tensile strength of the material was tested by a computer control electronic universal tester (Jinan Heng Rui Jin Testing Machine Co., Ltd., Jinan, China) according to ISO 1924-2:2008.

#### 2.3.2. Limiting Oxygen Index (LOI) Test

The LOI values for all paper samples were measured using an HC-2 type instrument ((Nanjing Jiangning Analytical Instrument Co., Ltd., Nanjing, China) based on GB/T 5454-1997.

#### 2.3.3. Vertical Flammability (VF) Test

The VF test was carried out according to GB/T 14656-2009. The composite papers to be tested were 70 mm × 150 mm. The paper was continuously ignited for 12 s with a 4 cm flame height. The humidity of the papers was adjusted according to GB/T 10739-2002 before the VF tests.

#### 2.3.4. Scanning Electron Microscopy (SEM) Analysis

The residues after the VF tests were investigated using a scanning electron microscope (SEM, Quanta 250 FEG, FEI, Hillsboro, OR, USA) at a beam voltage of 5.0 kV. Previously, the samples were coated with a conductive layer of platinum.

#### 2.3.5. Raman Spectroscopy Analysis

The Raman spectra of the residues after the VF tests were obtained from a Raman spectrometer (Thermo Scientific DXR2, Waltham, MA, USA) with a 785 nm excitation line with an Ar ion laser source.

#### 2.3.6. Cone Calorimetry (CONE) Analysis

According to the ISO Standard 5660-1, the combustion properties of the samples were analyzed using a Dual Cone Calorimeter (CONE, Fire Testing Technology, UK) at a heat flux of 35 kW/m^2^. The specimen (10 ± 0.1 g) size was 100 × 100 × 2 mm^3^.

#### 2.3.7. Thermogravimetric (TG) Analysis

TG analysis (TG209F3, NETZSCH Group, Selb, Germany) was performed from 50 to 800 °C at a heating rate of 20 °C/min in air. The mass of each sample was approximately 5 mg. The samples were dried in an electrothermal blast drying box at 70 °C for 6 h before testing.

#### 2.3.8. Thermogravimetric Analysis Coupled with Fourier Transformed Infrared Analysis (TG-FTIR) in N_2_

The thermal stability and gas emissions of FR-PA, Ca-Alg, and FR-PA/Ca-Alg (30/70) composite paper were investigated by TG analysis in a TG209F3 instrument (Germany) connected with a Nicolet iS 50 FTIR spectrometer, from 50 to 800 °C, in a N_2_ atmosphere with a heating rate of 20 °C/min and a N_2_ flow rate of 50 mL/min. The samples were dried in an electrothermal blast drying box at 70 °C for 6 h before testing.

## 3. Results and Discussion

### 3.1. Physical Properties of the Prepared Paper

The surface micromorphology of the paper samples was studied by SEM, and the photographs are shown in Appendix A. The surface of Ca-Alg and FR-PA/Ca-Alg composite papers are similar, indicating that FR-PA and Ca-Alg can be mixed evenly in the papermaking process. The thickness of the prepared papers is almost the same for samples with the same absolute dry pulp weight (see Appendix A). As shown in Appendix A, the surface properties of the prepared paper were studied. The Bendtsen roughness and air permeability of the prepared paper samples are comparable, which is consistent with the similar morphology observed from the SEM analysis. The high roughness of the prepared paper is due to the periodic dent on the surface of the paper caused by the mold in the papermaking process. Furthermore, as shown in Appendix A, the tensile index of the composite paper decreased with increasing FR-PA content, indicating that the introduction of FR-PA causes a slight reduction in the strength of the composite paper, but its strength was not related to its flame-retardancy properties as suggested by the subsequent VF tests.

### 3.2. Vertical Flammability (VF) and Limiting Oxygen Index (LOI) Analysis

VF and LOI are typically used to evaluate the flame-retardant properties of materials. Photographs of the samples captured at a certain time during VF tests are shown in Figure 2, and the related data are presented in Table 1. As shown in Figure 2, commercial wood pulp paper is very flammable, with an LOI value of 18.6%. The Ca-Alg paper cannot be ignited, but significant smoldering occurs with a long afterglow time of 540 s, resulting in complete damage to the sample. When the FR-PA content is 10%, the after-flame time of the composite paper is 0 s, and smoldering is self-extinguished although it still has a long after-flame time of 125 s. In addition, the damaged length (51 mm) is greatly reduced, indicating that the addition of FR-PA was beneficial to inhibit the smoldering of Ca-Alg. Furthermore, when the FR-PA content is 20% or 30%, although the after-flame time (0.4 s, 1.4 s) and damaged length (65 mm, 63 mm) of the composite paper are slightly increased, the afterglow time is significantly shortened (74 s and 53 s, respectively). Particularly, the FR-PA/Ca-Alg (30/70) composite paper meets the standard of flame-retardant paper, and FR-PA/Ca-Alg (20/80) composite paper is also very close to this standard. However, when the content of FR-PA is 40%, the composite paper combusted completely, which may be due to the excess heat generated by the excessive content of FR-PA. Specifically, as shown in Table 2, the combustion of FR-PA produces a considerable amount of heat: 27.76 MJ/m^2^. As shown in Table 1, the LOI values of FR-PA/Ca-Alg (10/90), FR-PA/Ca-Alg (20/80), and FR-PA/Ca-Alg (30/70) composite papers (30.3%, 28.7%, and 29.0%, respectively) were higher than those of pure Ca-Alg paper (27.3%), while that of the FR-PA/Ca-Alg (40/60) composite paper (25.5%) was lower. These results show that the addition of an appropriate amount of FR-PA can effectively inhibit the smoldering of Ca-Alg-based composite papers and improve their flame retardancy.

### 3.3. SEM Analysis

The SEM images of the composite papers after the VF tests are presented in Figure 3. When heated, Ca-Alg degrades easily and forms pores on the surface (Figure 3A) due to its low initial thermal degradation temperature. The resulting pores result in contact between oxygen and the incompletely degraded area inside the fiber, resulting in the smoldering of the pure Ca-Alg paper. As expected, when the content of FR-PA is 10–30% (Figure 3B–D), the pores on the surface of the Ca-Alg fiber residue decrease significantly or disappear due to a carbon layer of FR-PA covering the fiber surface. The carbon layer formed by FR-PA isolates the contact between oxygen and the inner layer of the materials, inhibiting the smoldering of Ca-Alg. On the contrary, when the FR-PA content is 40% (Figure 3E), the surface of the Ca-Alg fibers presents a porous morphology again, and adhesion occurs between the fibers, which may be because the excess heat generated by the combustion of FR-PA fiber makes the Ca-Alg fiber burn more fully.

In general, a compact and graphitic carbon layer can prevent mass/heat transfer between the gas and the condensed phase [37]. The carbonaceous char quality of the samples tested after VF was characterized by Raman spectroscopy. The peaks at 1568 cm^−1^ (G band) and 1302 cm^−1^ (D band) are caused by the first-order scattering of the E_2g_ phonon of the sp^2^ carbon atom and the activation of the first-order scattering process of the sp^3^ carbon atom, respectively. The graphitization degree of the char residues was evaluated by an integrated area ratio of the D and G bands (I_D_/I_G_). Low I_D_/I_G_ values of char translate into a high graphitization degree [37]. The char residue of FR-PA/Ca-Alg (20/80) shows the highest graphitization degree, followed by FR-PA/Ca-Alg (10/90) and FR-PA/Ca-Alg (30/70). On the other hand, the char residue of Ca-Alg and FR-PA/Ca-Alg (40/60) shows a lower graphitization degree. These results demonstrate that a proper FR-PA content can form a compact carbon layer on the surface of the Ca-Alg fibers during combustion.

As shown in Figure 3a–e, the peak at 1083 cm^−1^ is attributed to the symmetric C–O stretching mode for calcite, the peak at 711 cm^−1^ is ascribed to the O–C–O bending (in-plane deformation) mode for calcite, and the peak at 267 cm^−1^ is assigned to the libration mode of CO_3_ for calcite [38,39]. Calcite is the product of thermal degradation of Ca-Alg. It can be observed from Figure 3a–d that the intensity of the characteristic peak of calcite decreases with increasing FR-PA content, which also proves that the carbon layer formed by FR-PA can inhibit the further degradation of Ca-Alg. When the content of FR-PA is 40%, the intensity of calcite increases again, which is because the heat generated by excessive FR-PA combustion makes Ca-Alg degrade more completely and produce more CaCO_3_. These results show that the flame-retardant mechanism of the composite paper is of the condensed phase flame-retardant type, and confirms that the addition of a proper content of FR-PA can inhibit the smoldering of the Ca-Alg fibers.

### 3.4. CONE Analysis

CONE is an important testing method that can capture comprehensive information on the flame-retardancy performance of materials [40,41,42,43], such as time to ignition (TTI), pHRR, the peak of smoke production rate (pSPR), THR, and total smoke production (TSP), among others. The data obtained from this analysis are shown in Table 2 and Figure 4. It can be observed that the TTI of commercial wood pulp paper is only 17 s, and the pHRR (176.92 kW/m^2^) is reached in a short time (34 s). On the contrary, the TTI (58 s) of the Ca-Alg paper is more than three times that of wood pulp paper, and the pHRR (135.83 kW/m^2^) and THR of the Ca-Alg paper are lower than those of wood pulp paper, as well as the pSPR and TSP, which confirms that Ca-Alg fiber can potentially be used for the fabrication of flame-retardant paper. However, the smoldering of Ca-Alg paper cannot be neglected, as shown in the VF tests. Therefore, it is necessary to study the combustion behavior of the FR-PA/Ca-Alg composite paper. In the composite papers, TTI decreases with increasing FR-PA content (the TTI of FR-PA/Ca-Alg (10/90), FR-PA/Ca-Alg (20/80) and FR-PA/Ca-Alg (30/70) is 52 s, 50 s, and 37 s, respectively), because the FR-PA melts and covers the surface of the Ca-Alg before ignition, increasing the surface temperature of the Ca-Alg, further resulting in the composite papers being ignited ahead of time. Specifically, FR-PA/Ca-Alg (30/70) has two HRR peaks, which are located at 55 and 80 s, respectively. This phenomenon is due to the fact that the dense carbon layer formed by the melting of FR-PA weakens the transfer of heat to the inner layer of the material and blocks part of the volatiles from entering the combustion zone at the initial stage of combustion so that the HRR tends to decrease after the first peak.

The combustion of polymers is a complex process, so it is not rigorous to evaluate the flame retardancy of materials by a single parameter (such as TTI, pHRR, or THR). The fire growth index [44] (FGI), which is defined as the ratio of pHRR to t-pHRR, reflects the flame diffusion rate after the material is ignited. The larger the FGI, the easier the flame spreads and the greater the potential fire hazard of the material. The fire performance index [45] (FPI), that is, the ratio of TTI to pHRR, has been an important parameter to describe the flashover propensity of materials. The higher the value of FPI, the better the flame retardancy of the materials. In addition, Henri et al. proposed a simple and general dimensionless criterion, the flame-retardant index (FRI) [46], which can be used to quantify the flame retardancy of different polymer composites. The value of FRI in the range of “0–1”, “1–10”, “10–100” corresponds to “poor”, “good”, and “excellent” flame-retardant properties of the material, respectively. The FGI, FPI, and FRI values of the samples are shown in Figure 4e. Compared with commercial wood pulp paper, the FGI values of the composite papers are lower, the FPI values are higher, and the FRI values are in the “good” range. These excellent results show that the fire risk of composite paper is lower than that of commercial wood pulp paper.

Smoke is a non-negligible factor in fires and the main cause of death [47]. As shown in Figure 4b, the TSP of composite paper is higher than that of commercial wood pulp paper, which is caused by the addition of different contents of FR-PA. However, compared with FR-PA, the TSP of the composite papers FR-PA/Ca-Alg (10/90), FR-PA/Ca-Alg (20/80), and FR-PA/Ca-Alg (30/70) decreased by 88%, 87%, and 93%, respectively, indicating that Ca-Alg changed the combustion process of FR-PA and reduced smoke production. The effective heat of combustion (EHC) represents the heat released by the combustion of the components in the volatiles formed by the thermal decomposition of the material, which is helpful to analyze its flame-retardant mechanism. The average EHC (av-EHC) of the composite papers is lower than that of Ca-Alg paper, which may be due to the fact that the flame retardant contained in FR-PA inhibits the further combustion of volatile gas. Simultaneously, the amount of residue of the composite papers is higher than that of Ca-Alg paper, which indicates that FR-PA also plays a flame-retardant role in the condensed phase.

It is worth noting that carbon monoxide production (COP) within 100–300 s (Figure 4c) is caused by the smoldering of Ca-Alg, and the decreased peak intensity of the composite papers shows that the addition of FR-PA can effectively inhibit the smoldering of Ca-Alg. Carbon dioxide is produced (CO_2_P) both in the Ca-Alg and composite papers before ignition (Figure 4d), which is attributed to the decarboxylation of Ca-Alg which dilutes the concentration of the combustible gas and delays the ignition of the material.

Overall, compared with wood pulp paper, the FR-PA/Ca-Alg (30/70) composite paper has a lower FGI value and higher FPI and FRI values, indicating that composite paper has a lower fire risk. Compared with Ca-Alg paper, the smoldering of FR-PA/Ca-Alg (30/70) composite paper is self-extinguishing, and the afterglow time is 53 s, which reduces possible harm caused by smoldering.

### 3.5. TG Analysis in N_2_ and Air

The thermal stability of the composites in N_2_ was analyzed, and the results are shown in Figure 5 and Table 3. Specifically, the temperature of 5% weightlessness (T_5%_) is considered to be the initial degradation temperature of the materials [48]. The T_5%_ of FR-PA is 375 °C, indicating that the fibers have excellent heat resistance. In contrast, the T_5%_ of Ca-Alg is 104 °C, because of the hygroscopic behavior of the Ca-Alg fibers, which releases water vapor at the initial stage of heating to inhibit the ignition of fibers [18]. The thermal degradation of FR-PA is violent and occurs at 375–500 °C, with a maximum thermal degradation temperature (T_max-C_) of 427 °C, and final residual char of 3.99%. The thermal degradation of Ca-Alg can be divided into three stages: stage A (50–205 °C), where the weight loss is of approximately 12% attributed to the release of water in the fibers, stage B (205–600 °C), where the weight loss is approximately 41% and is attributed to the fracture of glycosidic groups, dehydration, and decarboxylation, and stage D (600–800 °C), where the weight loss is 6% due to the decomposition of calcium carbonate to form calcium oxide and calcium hydroxide [20].

Compared with the control samples, the thermal degradation process of the composite papers can be divided into four stages: stage A (50–205 °C), stage B (205–350 °C), stage C (350–600 °C), and stage D (600–800 °C). Stages A, B, and D are mainly attributed to the thermal degradation of Ca-Alg, and FR-PA plays no significant role. The weight loss rate at T_max_ (R_max_) is directly proportional to the content of Ca-Alg fibers in the sample. A small difference can be observed in stage C compared with the control sample, and T_max_s of the composite papers (for FR-PA/Ca-Alg (30/70) and FR-PA/Ca-Alg (20/80) it was 420 and 419 °C, respectively,) which occurred earlier than that of FR-PA (427 °C). However, R_max_ decreased, which may be attributed to the fact that the intermediate products of the thermal degradation of Ca-Alg slow down the degradation process of FR-PA.

The thermal oxidative degradation behavior of the samples in air was also studied. The TG and DTG curves and related data are shown in Figure 6 and Table 4, respectively. It is worth noting that the incorporation of FR-PA improved the thermal oxygen stability of the composite papers in the low-temperature range of 50 to 350 °C, which is due to the high thermal stability of FR-PA in this temperature zone. In addition, the thermal oxidative degradation of the composite paper was relatively mild in the medium temperature range (350–500 °C). The Ca-Alg fiber affected the thermal-oxidative degradation process of FR-PA and greatly reduced the R_max_ (at 429 °C) of FR-PA. This may be related to the fact that calcium ions in Ca-Alg fibers catalyze specific chemical reactions during thermal oxidative degradation, such as carbonization, dehydration, and cyclization, thus reducing the rate of thermal degradation [18,19,29,49]. In the high-temperature region (500–800 °C), the R_max_ (about at 552 °C) of composite paper was lower than that of Ca-Alg paper, T_max_s occurred earlier, and the char residues increased. Assuming that there is no interaction between Ca-Alg and FR-PA in the process of thermal oxidative degradation, the char residues of FR-PA/Ca-Alg (30/70) and FR-PA/Ca-Alg (20/80) composite papers should be 7.94% and 8.95% respectively, but the char residues of both were higher than their theoretical values. This phenomenon may be attributed to the fused carbon formed by FR-PA, which protects the Ca-Alg fiber from mass and heat transfer, further inhibiting the smoldering of Ca-Alg.

### 3.6. Infrared Spectrum Analysis of Gaseous Volatiles Acquired from TG Test

FTIR of the gaseous volatiles produced in the TG test of the specimens was conducted [50]. Three-dimensional diagrams of the FTIR absorption of the gases evolved for the thermal degradation are shown in Figure 7, and the FTIR spectra at T_max_s and other specific products are further compared in Figure 8a and b, respectively. As shown in Figure 8a, in the second stage of thermal degradation (205–350 °C, T_max_ is 259 °C) of FR-PA/Ca-Alg (30/70) composite paper, the formation of CO_2_ is attributed to the decarboxylation of Ca-Alg, while the FR-PA has no absorption peak at this stage, indicating that the FR-PA has no effect on the thermal degradation of Ca-Alg. Specifically, the bands at 2356 and 2313 cm^−1^ are attributed to CO_2_ vibrations. The CO_2_ formed at the initial stage of thermal degradation is beneficial to decrease the concentration of the combustible gas and inhibit the ignition of the composite paper.

In the third stage of thermal degradation (350–600 °C, T_max_ is 420 °C) of the FR-PA/Ca-Alg (30/70) composite paper, the peak at 2935 cm^−1^ is attributed to the vibration of aliphatic C–H groups, the bands at 1763 and 1706 cm^−1^ are ascribed to the C=O vibration of aldehydes and ketones, respectively, and the bands at 1625, 1505 and 1452 cm^−1^ are attributed to the C=C stretching vibration of the aromatic ring. In addition, the band at 1141 cm^−1^ is assigned to C–O–C vibration, while the peaks at 963 and 927 cm^−1^ are ascribed to NH_3_, which is a specific pyrolysis gas of FR-PA [29]. Compared with FR-PA, the peaks at 1625 and 1452 cm^−1^ decreased and the peak at 1763 cm^−1^ increased for the FR-PA/Ca-Alg composite paper, indicating that Ca-Alg changed the degradation process of FR-PA, reduced the formation of NH_3_ and aromatic products, and increased the formation of aldehyde products. It is possible that the Ca^2+^ ions from the Ca-Alg fibers catalyze the char formation of FR-PA [18,19,29,49], reduce the generation of volatile combustibles, and the resulting char deposited on the surface of the Ca-Alg fibers inhibits the smoldering of Ca-Alg. Moreover, the absorption intensity of hydrocarbons, ether components and NH_3_ for FR-PA/Ca-Alg are lower and the absorption intensities of CO_2_ are higher than those of FR-PA. According to the Lambert–Beer law [29], this means that the degradation of the composite paper produces more non-combustible gases and less combustible gases, which inhibits the spread of flame after ignition.

## 4. Conclusions

FR-PA/Ca-Alg composite papers with low flame-retardant content were prepared by blending. When the content of FR-PA is 30%, FR-PA effectively suppresses the smoldering of Ca-Alg and increases the LOI values. The afterglow time is 53 s, which meets the requirements of flame-retardant paper. SEM images and Raman spectroscopy confirmed that the molten FR-PA forms a compact carbon layer on the surface of the Ca-Alg, which isolates the mass/heat transfer and inhibits the smoldering of Ca-Alg. The CONE results showed that composite paper has a low fire risk (lower FGI value, and higher FPI and FRI values compared with Wood pulp paper). TG and TG-FTIR analysis indicated that Ca-Alg effectively inhibits the formation of aromatic products and NH_3_, and the CO_2_ produced by the degradation of Ca-Alg showed gas-phase flame-retardant activity on the combustion of composite paper. In addition, for FR-PA/Ca-Alg (30/70) paper, the flame-retardant content is only 1.5 wt%, and this low flame-retardant usage is in line with environmental protection efforts. Benefiting from this work, a method to inhibit the smoldering of Ca-Alg fibers is proposed and a new type of material is developed, which can be used as flame-retardant paper.

## Figures and Tables

**Figure 1 polymers-13-00430-f001:**
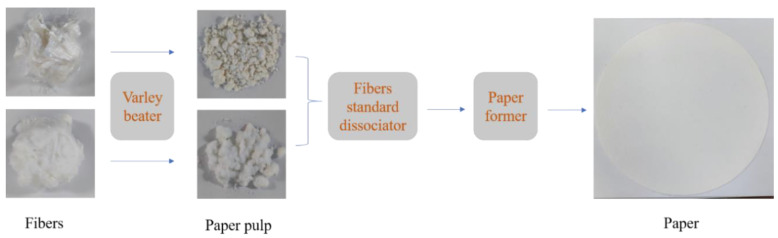
Preparation process of paper.

**Figure 2 polymers-13-00430-f002:**
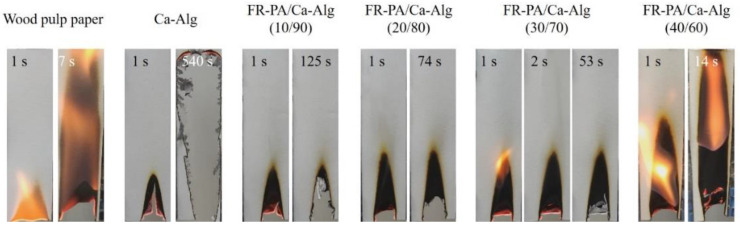
Digital photo of composite papers at different times during the Vertical Flammability (VF) tests. For all papers, the photos of the fire source leaving the sample for 1 s are shown; photos of the sample being completely destroyed are shown for wood pulp paper (7 s), Ca-Alg paper (540 s) and flame-retardant polyamide (FR-PA)/Ca-Alg (40/60) paper (14 s); and photos after smoldering is extinguished are shown for FR-PA/Ca-Alg (10/90) paper (125 s), FR-PA/Ca-Alg (20/80) paper (74 s), and FR-PA/Ca-Alg (30/70) paper (53 s). In addition, the photo after the flame of the paper is extinguished is shown for FR-PA/Ca-Alg (30/70) paper (2 s).

**Figure 3 polymers-13-00430-f003:**
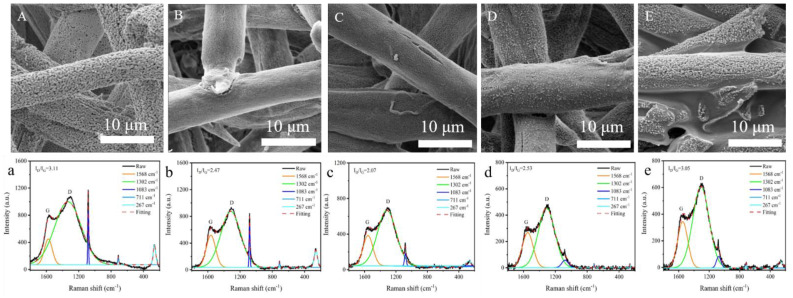
SEM images of composite paper after VF test: (**A**) Ca-Alg, (**B**) FR-PA/Ca-Alg (10/90), (**C**) FR-PA/Ca-Alg (20/80), (**D**) FR-PA/Ca-Alg (30/70), and (**E**) FR-PA/Ca-Alg (40/60); Raman spectra of char residues of (**a**) Ca-Alg, (**b**) FR-PA/Ca-Alg (10/90), (**c**) FR-PA/Ca-Alg (20/80), (**d**) FR-PA/Ca-Alg (30/70), and (**e**) FR-PA/Ca-Alg (40/60) after VF test.

**Figure 4 polymers-13-00430-f004:**
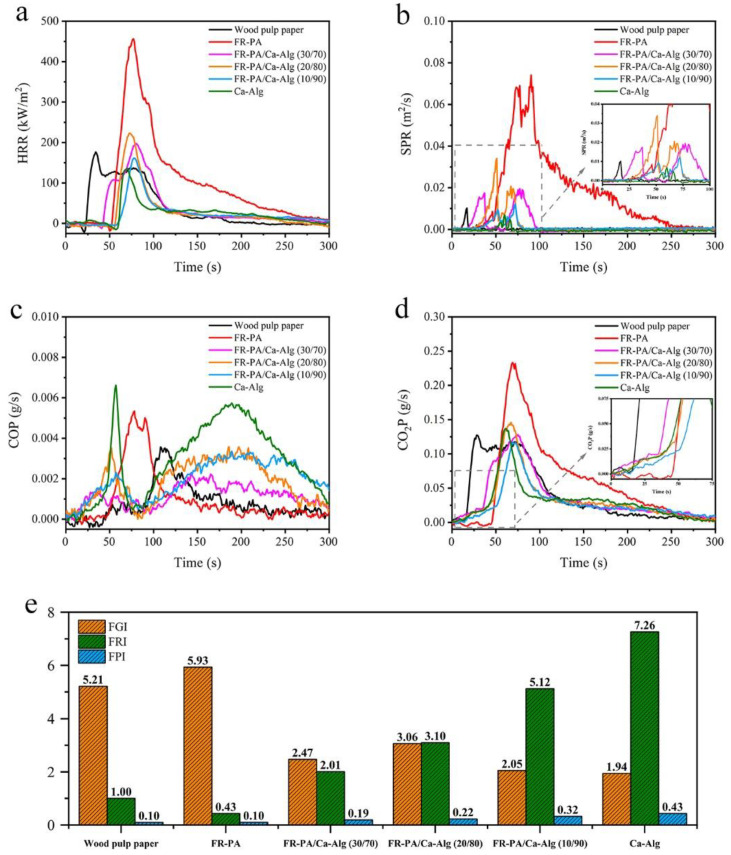
(**a**) Heat release rate (HRR), (**b**) smoke production rate (SPR), (**c**) carbon monoxide production (COP) and (**d**) CO_2_P of samples acquired from CONE; (**e**) the fire growth index (FGI), fire performance index (FPI) and flame-retardant index (FRI) values of samples.

**Figure 5 polymers-13-00430-f005:**
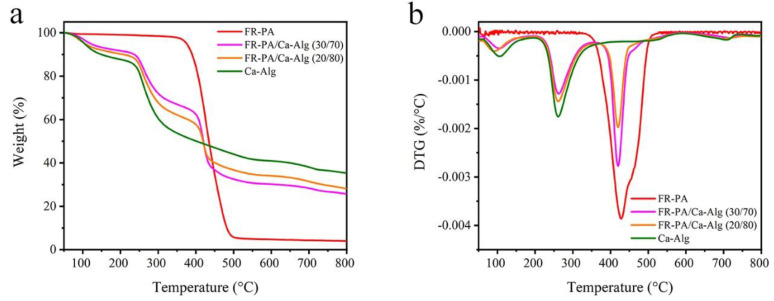
Thermogravimetric (TG) (**a**) and differential thermogravimetric (DTG) (**b**) curves in N_2_ for samples.

**Figure 6 polymers-13-00430-f006:**
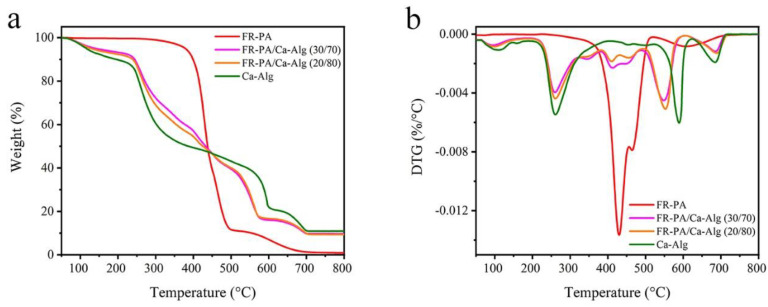
TG (**a**) and DTG (**b**) curves in air for samples.

**Figure 7 polymers-13-00430-f007:**
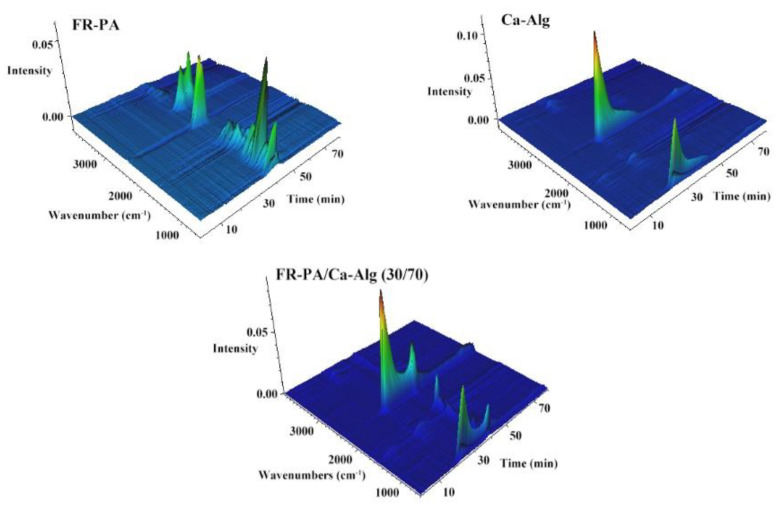
3D diagrams of FTIR absorption of the thermal degradation of specimens.

**Figure 8 polymers-13-00430-f008:**
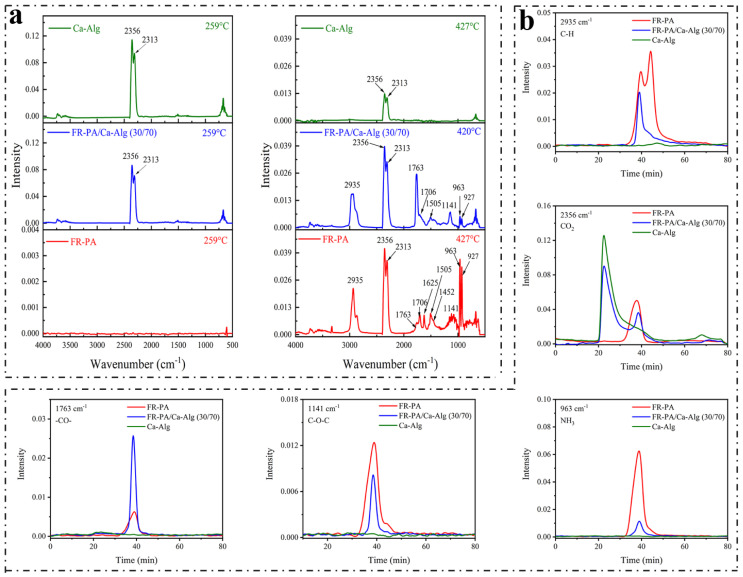
(**a**) FTIR spectra at T_max_ of specimens studied; (**b**) several specific products versus time are compared.

**Table 1 polymers-13-00430-t001:** Data corresponding to the limiting oxygen index (LOI) and VF tests.

Samples	LOI (%)	After-Flame Time ^a^ (s)	Afterglow Time ^a^ (s)	Damaged Length ^a^ (mm)
Wood pulp paper	18.5 (0.17) ^b^	7.5 ± 1	-	All damaged
Ca-Alg	27.3 (0.17)	0	540 ± 10	All damaged
FR-PA/Ca-Alg (10/90)	30.3 (0.17)	0	125 ± 5	51 ± 5
FR-PA/Ca-Alg (20/80)	28.7 (0.28)	0.4 ± 0.2	74 ± 6	65 ± 4
FR-PA/Ca-Alg (30/70)	29.0 (0.22)	1.4 ± 0.3	53 ± 5	63 ± 4
FR-PA/Ca-Alg (40/60)	25.5 (0.35)	14.2 ± 0.6	-	All damaged

^a^: Average value of four repeated tests. ^b^: The values in the parenthesis indicate standard deviation.

**Table 2 polymers-13-00430-t002:** Parameters acquired from cone calorimetry (CONE) of specimens.

Samples	TTI (s)	pHRR (kW/m^2^)	t-_pHRR_ (s)	THR (MJ/m^2^)	TSP (m^2^/m^2^)	av-EHC (MJ/kg)	Residue (%)
Wood pulp paper	17	176.92	34	11.63	0.11	12.31	20.90
FR-PA	45	456.54	77	27.76	4.92	27.12	3.35
FR-PA/Ca-Alg (30/70)	37	197.31	80	11.27	0.57	15.06	13.25
FR-PA/Ca-Alg (20/80)	50	223.17	73	8.77	0.61	14.21	14.08
FR-PA/Ca-Alg (10/90)	52	161.63	79	7.61	0.32	12.16	14.98
Ca-Alg	58	135.83	70	7.12	0.05	16.53	12.15

**Table 3 polymers-13-00430-t003:** The TG and DTG data in N_2_ of specimens.

Samples	T_5%_ (°C)	T_max−A_ (°C)	R_max−A_ (%/°C)	T_max−B_ (°C)	R_max−B_ (%/°C)	T_max−C_ (°C)	R_max−C_ (%/°C)	T_max−D_ (°C)	R_max−D_ (%/°C)	W _f_ (%)
FR-PA	375	-	-	-	-	427	−0.0039	-	-	3.99
FR-PA/Ca-Alg (30/70)	120	103	−0.0003	259	−0.0013	420	−0.0028	720	−0.0001	25.72
FR-PA/Ca-Alg (20/80)	105	104	−0.0004	259	−0.0014	419	−0.0019	719	−0.0001	28.20
Ca-Alg	104	103	−0.0005	259	−0.0018	-	-	712	−0.0001	35.34

**Table 4 polymers-13-00430-t004:** The TG and DTG data in air of specimens.

Samples	T_5%_ (°C)	T_max−1_ (°C)	R_max−1_ (%/°C)	T_max−2_ (°C)	R_max−2_ (%/°C)	T_max−3_ (°C)	R_max−3_ (%/°C)	T_max−4_ (°C)	R_max−4_ (%/°C)	T_max−5_ (°C)	R_max−5_ (%/°C)	W _f_ (%)
FR-PA	379	-	-	-	-	429	−0.0137	607	−0.0008	-	-	0.89
FR-PA/Ca-Alg (30/70)	141	94	−0.0007	259	−0.0040	412	−0.0023	550	−0.0045	687	−0.0012	9.64
FR-PA/Ca-Alg (20/80)	128	98	−0.0008	260	−0.0044	412	−0.0018	552	−0.0051	689	−0.0013	9.33
Ca-Alg	116	109	−0.0011	260	−0.0055	-	-	589	−0.0060	685	−0.0019	10.96

## Data Availability

The data presented in this study are available on request from the corresponding author.

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
