# Peer review of "Suppression of Smoldering of Calcium Alginate Flame-Retardant Paper by Flame-Retardant Polyamide-66"

_polymers, 2021, doi:10.3390/polym13030430_

Round 1

Reviewer 1 Report

This work reported the suppression of smoldering of calcium alginate flame-retardant paper by blending flame-retardant polyamide-66. The work is well organized and comprehensive. The conclusions are supported by the results. Some minor issues related to this work should be considered before publication.

1) What is the FR PA fiber? Please provide more details about the fiber which enables researchers to repeat the experiment.

2) What about the mechanical property of the final composite paper?

3) Only SEM is not enough to investigate the char residues for the flame retardant paper. Please further investigate the structure of the char residues by Raman, FTIR…. (Chemical Engineering Journal. 2020;399:125829)

4) For cone calorimetry and TG-IR tests, some important references are missing (Journal of Hazardous Materials. 2021;401:123342; Journal of Materials Chemistry A. 2016;4(19):7330-40).

In addition, how to obtain the absorption intensity of various gaseous products (e.g. per mg) from TG-IR results?

Author Response

Point 1: What is the FR-PA fiber? Please provide more details about the fiber which enables researchers to repeat the experiment.

Response 1:

The detailed information of FR-PA is supplemented in the Materials (line 89). FR-PA fibers were supplied by Huafeng Group Co., Ltd. (China). The flame retardant contained in FR-PA is phosphorus-based flame retardant, with a 5 wt% content, and the structural formula is shown in Figure S1.

Point 2: What about the mechanical property of the final composite paper?

Response 2:

For wood pulp paper, Ca-Alg paper and composite paper, absolutely dry pulp (10g) is used to prepare paper by the Rapid-Kothen method (Biomacromolecules 2010, 11, 2195–2198). In the case of the same absolute dry pulp weight, the thickness of wood pulp paper, Ca-Alg paper and composite paper are the same. In addition, we conducted related tests on the tensile properties of Ca-Alg paper and composite paper. The mechanical properties of Ca-Alg paper and composite paper are not much different (line 166), and their mechanical properties are not related to combustion properties.

Point 3: Only SEM is not enough to investigate the char residues for the flame retardant paper. Please further investigate the structure of the char residues by Raman, FTIR…. (Chemical Engineering Journal. 2020;399:125829).

Response 3:

The reference (Chemical Engineering Journal. 2020; 399: 125829) have given us important help. The char residues of paper were analyzed by Raman spectroscopy (line 178). A proper amount of FR-PA can form a compact carbon layer on the surface of the Ca-Alg to prevent mass/heat transfer and inhibit the smoldering of Ca-Alg. When the FR-PA content is higher, the excessive heat generated by FR-PA combustion (CONE analysis shows that FR-PA combustion generates a lot of heat) destroys the carbon layer, makes the degradation of Ca-Alg more completely.

Point 4: For cone calorimetry and TG-IR tests, some important references are missing (Journal of Hazardous Materials. 2021;401:123342; Journal of Materials Chemistry A. 2016;4(19):7330-40).

Response 4:

The missed references (reference 41 and 48) are very helpful to our paper, thank you very much for your help.

Point 5: How to obtain the absorption intensity of various gaseous products (e.g. per mg) from TG-IR results?

Response 5:

TG-IR analysis were carried out by a TG analyzer coupled with a FTIR spectrophotometer. The gas products released from the pyrolysis in TG were swept immediately to a cold gas cell for analysis by FTIR. The samples to be tested approximately 10 mg. According to the widely used Lambert–Beer law (Journal of Analytical and Applied Pyrolysis. 100 (2013) 26-32), the absorption spectrum at a specific wave number is linearly dependent on gas concentration. Thus, the variation of absorbance in the whole pyrolysis process reflects the concentration trend of the gas species.

Reviewer 2 Report

The manuscript described the preparation and characterization of flame-retardant papers based on composites made of polyamide and calcium alginate. The text is clear and concise, with most part of the results well presented and discussed. Based on this, I support its publication in Polymers with minor changes, as stated below:

As a general curiosity, can a material with absence of wood pulp on its composition be called 'paper'? The composites investigated are based on a blend of a synthetic and a natural polymer, so it would be interesting to state if such a polymeric composite can be classified as a wood-based material.

Data displayed in Fig. 2 (caption says Fig. 1 - please, correct that) is very interesting, but difficult to be compared between each other because the time intervals are very different. Could the authors provide images in the same, or similar, time intervals?

EDS analysis based on SEM images could corroborate the formation of a carbon layer in some of the composites, especially on those with higher FR-PA content. Can the authors provide such data based on the images already recorded?

Also about the SEM images, do they have enough resolution, in addition to a sufficient number of images, that enable the obtention of a size distribution histogram regarding the pore sizes?

Is there any previous work that shows the catalytic effect of calcium ions on the thermal degradation of polymeric materials? If so, such a reference should be included (perhaps the ref 14 and 15 of the manuscript). If not, a comment in this sense should be present in the manuscript.

Can any quantitative information, in terms of products formed by the thermal degradation, be extracted from FTIR data? In the end of the text, authors mention very briefly about the intensities of the bands related to some products, but, if possible, quantitative data would strengthen such discussion.

As a final comment, information about costs of production of the investigated composites, compared to other flame-retardant materials already available in the market, should be included in the new version of the manuscript.

Author Response

Point 1: As a general curiosity, can a material with absence of wood pulp on its composition be called 'paper'? The composites investigated are based on a blend of a synthetic and a natural polymer, so it would be interesting to state if such a polymeric composite can be classified as a wood-based material.

Response 1:

As a renewable resource from the ocean, Ca-Alg fiber is prepared by wet spinning of sodium alginate which is a biological polysaccharide extracted from brown algae. It is feasible to call this kind of biopolysaccharide-based composite material as ‘paper’.

Point 2: Data displayed in Fig. 2 (caption says Fig. 1 - please, correct that) is very interesting, but difficult to be compared between each other because the time intervals are very different. Could the authors provide images in the same, or similar, time intervals?

Response 2:

The smoldering rate of Ca-Alg is very slow, Ca-Alg paper is completely destroyed in 540 s, while wood pulp paper and FR-PA/Ca-Alg (40/60) paper are completely destroyed in 7 s and 14 s, respectively. The afterflame time and afterglow time of different samples are also different. Therefore, providing images at the same time interval is not conducive to a comprehensive understanding the combustion process of the sample. We have provided the photos of the sample when fire source leaving the sample for 1 s, the flame goes out, and smoldering self-extinguishing or the sample is completely destroyed, respectively. For ease of understanding, a detailed explanation is added to the selected photos in Figure 2 (line 155).

Point 3: EDS analysis based on SEM images could corroborate the formation of a carbon layer in some of the composites, especially on those with higher FR-PA content. Can the authors provide such data based on the images already recorded?

Response 3:

EDS analysis based on SEM images was not performed due to the test conditions in our laboratory. But in order to further study the carbon layer in the residue, we analyzed the residue after the VF test by Raman spectroscopy (line 178). The proper content of FR-PA is beneficial to form a compact carbon layer on the surface of the Ca-Alg fiber and avoid further decomposition of the Ca-Alg fiber.

Point 4: Also about the SEM images, do they have enough resolution, in addition to a sufficient number of images, that enable the obtention of a size distribution histogram regarding the pore sizes?

Response 4:

As shown in Raman analysis, composite paper and Ca-Alg paper will generate CaCO3 when burned, and the generated CaCO3 will affect the pore size distribution on the fiber surface. Raman analysis has shown that lower content of FR-PA can form a compact carbon layer on the surface of Ca-Alg fibers. Therefore, we believe that the histogram of the pore size distribution is negligible.

Point 5: Is there any previous work that shows the catalytic effect of calcium ions on the thermal degradation of polymeric materials? If so, such a reference should be included (perhaps the ref 14 and 15 of the manuscript). If not, a comment in this sense should be present in the manuscript.

Response 5:

 Thank you very much for your reminder, the relevant references have been added (reference 18, 19, 29, 47). Previous work has shown that calcium ions have a catalytic effect on the thermal degradation of polymeric materials, such as alginate fiber, viscose fiber and sulfonated polyoxadiazole fiber.

Point 6: Can any quantitative information, in terms of products formed by the thermal degradation, be extracted from FTIR data? In the end of the text, authors mention very briefly about the intensities of the bands related to some products, but, if possible, quantitative data would strengthen such discussion.

Response 6:

TG-IR analysis were carried out by a TG analyzer coupled with a FTIR spectrophotometer. The gas products released from the pyrolysis in TG were swept immediately to a cold gas cell for analysis by FTIR. The samples to be tested approximately 10 mg. According to the widely used Lambert–Beer law (Journal of Analytical and Applied Pyrolysis. 100 (2013) 26-32), the absorption spectrum at a specific wave number is linearly dependent on gas concentration. Thus the variation of absorbance in the whole pyrolysis process reflects the concentration trend of the gas species.

Point 7: Information about costs of production of the investigated composites, compared to other flame-retardant materials already available in the market, should be included in the new version of the manuscript.

Response 7:

As a kind of renewable fiber from ocean, the extraction process of Ca-Alg fiber raw material leads to its cost higher than terrestrial renewable fiber. But compared with terrestrial renewable fiber, Ca-Alg fiber has the advantage of inherent flame retardant. The introduction of FR-PA can effectively inhibit the smoldering of Ca-Alg. It is feasible to apply FR-PA/Ca-Alg composite paper with low flame retardant content in the flame retardant field.

Reviewer 3 Report

In this work “Suppression of smoldering of calcium alginate flame-retardant paper by flame-retardant polyamide-66” the author has used a flame retardant polymer (polyamide) to suppress the smoulding of a flame retardant materials (calcium alginate) by simple blending. The author has casted the composite material by simple casting to make it like paper thick. They characterized the material using TGA, Come calorimeter, FTIR, SEM, VF and LOI. The work is elaborative characterized. However, few question which comes to my concern is given below as my comments:

Q1. The thickness of the final casted composites film was not mentioned which has a great role and may change the properties with respect to the thickness of the paper.Is the thickness of the composites paper uniform and even all over?

Q2. The most important characterization related to this study is the LOI and vertical flammability test. However, I don’t really see a significant improvement in the VF test result in the composite syste.. The minimal improvement is coming under the standard deviation of error since the author has mentioned the thickeness of the paper composites. Also the 30/70 component, a clear flame was visible in the sample in the VF test result digital image as well as in the 40/60 sample. Similarly the LOI test also indicates similar results with least improvement that can be considered as an improvement.

Q3. If we consider correlating the result of the TGA with VF, The TGA also doesn’t show any improvement or might be slight improvement in line with the VF, LOI test result.

Q4. The SEM images, why the pores again appears at higher concentration of PA is not clear to me. The explanation doesn’t sounds logical.

But I would say the author as comprehensively characterized the study with characterization really matches well with the objective of the study. Unfortunately, the paper cant be accepted with the fact that the author has comprehensively characterized and organized well the  work. Therefore, I have no choice but recommend to reject this paper.

Author Response

Point 1: The thickness of the final casted composites film was not mentioned which has a great role and may change the properties with respect to the thickness of the paper.Is the thickness of the composites paper uniform and even all over?

Response 1:

Composite paper is prepared by Rapid-Kothen method (Biomacromolecules 2010, 11, 2195–2198). At the same amount of absolute dry pulp (10g), the thickness of the composite paper prepared is the same and uniform. Furthermore, the micromorphology of Ca-Alg paper and composite paper was analyzed by SEM, and the morphology of different samples is not much different, indicating that FR-PA and Ca-Alg can be mixed evenly in the papermaking process..

Point 2: The most important characterization related to this study is the LOI and vertical flammability test. However, I don’t really see a significant improvement in the VF test result in the composite syste. The minimal improvement is coming under the standard deviation of error since the author has mentioned the thickeness of the paper composites. Also the 30/70 component, a clear flame was visible in the sample in the VF test result digital image as well as in the 40/60 sample. Similarly the LOI test also indicates similar results with least improvement that can be considered as an improvement.

Response 2:

For the selected pictures in Figure 2, we have added a detailed explanation (line 155). In the VF tests, wood pulp paper and FR-PA/Ca-Alg (40/60) paper were completely destroyed by afterflame, while Ca-Alg paper was completely destroyed by smoldering. When the content of FR-PA is 10 - 30%, the smoldering of the composite paper will self-extinguish. As the content of FR-PA increases, the afterglow time is gradually shortened. Although the afterflame time of FR-PA/Ca-Alg paper (1.4 s) is greater than that of Ca-Alg paper (0 s), it still meets the requirements of flame retardant paper (afterflame time is less than or equal to 5 s). The FR-PA/Ca-Alg (40/60) paper is completely damaged due to excessive heat generated by the combustion of FR-PA.

Therefore, the addition of an appropriate amount of FR-PA can effectively improve the flame retardant properties of composite paper and inhibit the smoldering of Ca-Alg.

Point 3: If we consider correlating the result of the TGA with VF, The TGA also doesn’t show any improvement or might be slight improvement in line with the VF, LOI test result.

Response 3:

Firstly, VF are used to characterize the flame retardant properties of composite paper. It can be observed from Figure 2 that the smoldering of FR-PA/Ca-Alg (30/70) composite paper self-extinguished at 53 s, while the Ca-Alg paper was completely destroyed due to smoldering, indicating that the introduction of FR-PA can inhibit the smoldering of Ca-Alg effectively.

Secondly, the thermal degradation performance of composite paper was studied by TG. The TG curve shows the residual mass of the sample as a function of temperature. Under N2 atmosphere, the thermal degradation process of FR-PA and Ca-Alg are almost independent of each other; therefore, the composite paper TG curve is almost a superposition of that for FR-PA and Ca-Alg. Under air atmosphere, the degradation process of composite paper (about before 400℃) is similar to that in a N2 atmosphere, indicating that the initial thermal degradation of Ca-Alg and FR-PA is non-oxidative degradation; as the temperature increases, the volatile components generated by degradation are ignited, which causes the sample to be ignited, and the residual amount of the sample decreases rapidly.

In short, in VF tests, FR-PA improves the flame retardancy of composite paper mainly in smoldering which occurs at a lower temperature; while TG reflects the change curve of sample weight with temperature, both Ca-Alg and FR-PA are completely degraded at 800℃. Therefore, There is almost no direct correlation between the test results of VF and TG,

Point 4: The SEM images, why the pores again appears at higher concentration of PA is not clear to me. The explanation doesn’t sounds logical.

Response 4:

The appearance of pores means that the thermal degradation of calcium alginate fiber is more complete. The compact carbon layer formed by FR-PA covers the surface of the Ca-Alg to isolate the mass/heat transfer and inhibit the smoldering of the Ca-Alg. When the FR-PA content is higher, the excessive heat generated by FR-PA burning is fed back to the Ca-Alg, which leads to more complete degradation of the Ca-Alg, the pores on the surface appears again.

Reviewer 4 Report

The authors carried a study about flame retardant system, my comments are below; 1. The introduction part should talk about the need of this study, add more recent references. 2. It would be better authors add some sentences on the consumption of fire resistance in different fields. Add the following paper for some stats of fire resistance (Global consumption of flame retardants and related environmental concerns: A study on possible mechanical recycling of flame retardant textiles). 3. In terms of formaldehyde resin, is it being replaced in your system? How it affects the burning behavior. Add this reference on how flame retardant work (Process optimization of eco-friendly flame retardant finish for cotton fabric: A response surface methodology approach). 4. The TGA in air and nitrogen show similar behavior, especially for FR-PA but the FTIR showed different totally different functional groups. Can you add comments on this? 5. The conclusion part should exclusively talk about the findings.

Author Response

Point 1: The introduction part should talk about the need of this study, add more recent references.

Response 1:

Thank you very much for your suggestions, we revised the introduction. Some examples of the flame retardant mechanism of flame retardants and the method of preparing flame retardant paper are supplemented (line 36).

Point 2: It would be better authors add some sentences on the consumption of fire resistance in different fields. Add the following paper for some stats of fire resistance (Global consumption of flame retardants and related environmental concerns: A study on possible mechanical recycling of flame retardant textiles).

Response 2:

Reference 15 help us understand the hazards of flame retardants to the environment, reducing the application of flame retardants, and strengthening the research on flame retardant recycling is imperative.

Point 3: In terms of formaldehyde resin, is it being replaced in your system? How it affects the burning behavior. Add this reference on how flame retardant work (Process optimization of eco-friendly flame retardant finish for cotton fabric: A response surface methodology approach).

Response 3:

For the Ca-Alg-based flame-retardant composite paper, FR-PA fiber, as one of the additives we considered, has an excellent effect in inhibiting the smoldering of Ca-Alg. The carbon layer formed by FR-PA covers the surface of the Ca-Alg, inhibiting the smoldering of the Ca-Alg. For other additives, we have also made some attempts, some of which also inhibited the smoldering of Ca-Alg, but it is not convenient for us to disclose it temporarily. Reference 12 help us better understand the working mechanism of flame retardants

Point 4: The TGA in air and nitrogen show similar behavior, especially for FR-PA but the FTIR showed different totally different functional groups. Can you add comments on this?

Response 4:

About before 400℃, the TG curves under N2 and air are similar, which due to the fact that the initial thermal degradation of FRPA and Ca-Alg is non-oxidative degradation, and oxygen does not work. As the temperature increases, the sample under the air atmosphere is ignited, and the TG curves under the nitrogen and air atmosphere begin to appear different.

TG-IR mainly shows the IR spectrum of the gaseous product generated by the pyrolysis of the sample, while the TG curve reflects the change in the weight of the sample in the condensed phase with temperature. The difference in functional groups in the gaseous product is due to the catalysis of Ca2+ changing the pyrolysis products of FR-PA, but it does not significantly affect the thermal weight loss process.

Point 5: The conclusion part should exclusively talk about the findings.

Response 5:

Thank you for your suggestions, we have re-summarized the conclusion and highlighted our findings.(line 324)

Round 2

Reviewer 3 Report

I am not satisfied with the authors explanation. The paper is of least interest to the reader and doesbt contribute fairly to the scientific side. Hence i recommend to reject this work in this journal.

Author Response

We have added information about thickness uniformity (as shown in Figure S3a), the thickness of the prepared paper is uniform and almost the same.

In this paper, FR-PA/Ca-Alg composite paper was prepared by blending and its flame retardancy was studied. The flame retardant contained in FR-PA induces FR-PA to form a compact carbon layer on the surface of the Ca-Alg fibers during combustion, the resulting carbon layer isolates the mass/heat transfer and effectively suppresses the smoldering of Ca-Alg, which belongs to the condensed flame-retardant mechanism. Furthermore, the combustion and thermal degradation behavior of composite paper were explained.

The language of the paper has been polished by native speakers, and the editing certificate is detailed in the attachment.

At present, there is a lack of research on inhibiting the smoldering of Ca-Alg, and we believe that this paper will attract the attention of reader. We are very much looking forward to your recognition of our work.

Reviewer 4 Report

The authors have answered all the comments.

Author Response

Dear reviewer

      The language of the paper has been polished by native speakers, and the editing certificate is detailed in the attachment.
